# Temozolomide Chronotherapy in Glioma: A Systematic Review

**Jason L. Jia** [1] , **Bader Alshamsan** [2,3] **and Terry L. Ng** [3,4,*]

¹ Core Internal Medicine Residency Program, Department of Medicine, University of Ottawa, Ottawa, ON K1H 8L6, Canada

² Department of Medicine, College of Medicine, Qassim University, Buraydah P.O. Box 6655, Saudi Arabia

³ Division of Medical Oncology, Department of Medicine, The Ottawa Hospital Cancer Centre, Ottawa, ON K1H 8L6, Canada

⁴ Cancer Therapeutics Program, Ottawa Hospital Research Institute, Ottawa, ON K1H 8L6, Canada

\* Correspondence: teng@toh.ca

**Abstract:** Outcomes for patients with high-grade glioma remain poor. Temozolomide (TMZ) is the only drug approved for first-line treatment of glioblastoma multiforme, the most aggressive form of glioma. Chronotherapy highlights the potential benefit of timed TMZ administration. This is based on pre-clinical studies of enhanced TMZ-induced glioma cytotoxicity dependent on circadian, oscillating expression of key genes involved in apoptosis, DNA damage repair, and cell-cycle mediated cell death. The current systematic review's primary aim was to evaluate the efficacy and toxicity of TMZ chronotherapy. A systemic review of literature following PRISMA guidelines looking at clinical outcomes on TMZ chronotherapy on gliomas was performed. The search in the English language included three databases (PubMed, EMBASE, and Cochrane) and five conferences from 1946 to April 2022. Two independent reviewers undertook screening, data extraction, and risk-of-bias assessment. A descriptive analysis was conducted due to limited data. Of the 269 articles screened, two unique studies were eligible and underwent abstraction for survival and toxicity findings. Both studies—one a retrospective cohort study (n = 166) and the other a prospective randomized feasibility study (n = 35)—were conducted by the same academic group and suggested a trend for improved overall survival, but possibly increased toxicity when TMZ was administered in the morning (vs. evening). There was limited evidence suggesting possible therapeutic value from administering TMZ in the morning, which may be consistent with the pre-clinical observations of the importance of the timing of TMZ administration in vitro. Larger, pragmatic, prospective randomized controlled trials are needed to ascertain the value of TMZ chronotherapy to provide optimized and equitable care for this population.

**Keywords:** chronotherapy; temozolomide; glioma; glioblastoma; chemotherapy; oncology





## 1. Introduction

Glioblastoma multiforme (GBM) is the most common form of adult glioma and is the most aggressive subtype due to its infiltrative growth pattern and heterogeneous biology [1]. Without any intervention (e.g., surgery, radiation, chemotherapy), the life expectancy from the diagnosis is less than three months [2]. The chance of long-term survival after treatment for GBM is <10% [3]. The first-line treatment for a newly diagnosed glioma, irrespective of histologic grade, involves temozolomide (TMZ) [4]. TMZ is an oral alkylating agent with good CNS penetration; it is one of the few drugs that has demonstrated a meaningful survival advantage in this population [2]. For adjuvant treatment of GBM specifically, TMZ is the sole preferred chemotherapeutic agent [4].

The current standard of treatment for newly diagnosed GBM was established in 2005, whereby the addition of TMZ to post-operative radiation demonstrated a 2.5-month improvement in median survival, leading to a median overall survival (OS) of 14.6 months or a two-year survival of 26% [5]. Despite over two decades of research, the standard of

treatment for gliomas has not changed significantly. Although the use of tumour-treating fields has demonstrated a clinically significant improvement in median OS [6], it is costly (USD 20,000/month) and is an externally worn device that relies on high levels of treatment adherence (≥18 h of wear per day). It has an evident impact on an individual's quality of life (e.g., the stigma of wearing a visible scalp device, scalp irritation, and discomfort), making it an intervention that is not broadly implementable into practice. More importantly, most patients relapse within the first two years of diagnosis, and there have been no proven effective second-line therapies after disease relapse [7].

There are inherent challenges to completing practice-changing studies for GBM. First, there is a relatively low incidence of these tumors. Second, the rapid clinical deterioration in many patients early on after surgery precludes them from entering rigid industry-sponsored studies that offer novel therapeutics. Third, there are inherent biological features (heterogeneous tumor biology, heavily immunosuppressed tumor microenvironment) of GBM that make it challenging to find a one-size-fits-all approach to treat this disease. While we wait for more novel and effective therapies, a more pragmatic approach to optimizing TMZ in GBM needs to be explored.

Chronotherapy refers to strategic medication timing to enhance existing therapies' benefits [8]. The expression of genes involved in cellular replication, metabolism, and DNA repair oscillate throughout the day, following a circadian rhythm; and are guided by the hypothalamic suprachiasmatic nucleus, which is subject to environmental cues such as light [9]. GBM cells appear to abide by this overarching temporal rhythm. They are known to express core clock genes basic helix-loop-helix ARNT like 1 (*bmal1*) and period circadian regulator 2 (*per2*) in a circadian fashion, and their deregulation is associated with glioma genesis and more aggressive behavior [10]. O6-methylguanine-DNA methyltransferase (MGMT), the protein repairing DNA damage induced by alkylating agents such as TMZ, also exhibits regular 5-fold variation oscillation of activity with a peak during night time and low light as determined in mouse liver [11]. TMZ-induced apoptosis is enhanced during peak *bmal1* expression, which appears to peak consistently approximately 5 h after dusk in post-mortem samples [12]. Chronotherapy has already shown some promise in leukemia and colorectal cancer [13,14].

As a result, leveraging these molecular temporal mechanisms can clinically improve existing therapeutics such as TMZ in GBM. TMZ represents an ideal chronotherapeutic agent for GBM. It is rapidly absorbed, readily crosses the blood–brain barrier, and exhibits a half-life of approximately 1.8 h [15]. Conceivably, if peak drug exposure can be synchronized to peak *bmal1* expression and nadir MGMT expression, then cytotoxicity, tumor control, and ultimately survival with the existing Stupp protocol may be improved.

Given the novelty of chronotherapy in this space, no evidence-based guidelines or statements address the value of, or lack thereof, the timing of TMZ administration to optimize survival and minimize treatment-related toxicity. If changing the timing of TMZ administration could significantly improve outcomes, then the implications to patient care would be massive. Therefore, we conducted a systematic review to summarize the currently available clinical evidence on the use of TMZ chronotherapy in the treatment of glioma.

## 2. Study Methods

### 2.1. Research Question and Study Eligibility Criteria

The research question for this study was: "What are the benefits and harms of temozolomide administered consistently at a particular time of day in patients with glioma?". We used the population-intervention-comparator-outcomes (PICO) framework to identify relevant studies for this review. The population described in eligible studies included patients with glioma (any WHO grade) receiving temozolomide (TMZ) during primary concurrent chemoradiotherapy, maintenance, or salvage therapy. The intervention/comparator was the timing (morning or evening) of TMZ administration. Pre-specified outcomes related to TMZ efficacy included overall survival (OS); progression-free survival (PFS); objective response rate (ORR); OS at 6, 9, and 12 months; PFS at 6, 9, and 12 months. Pre-specified

outcomes related to TMZ toxicity followed the Common Terminology Criteria for Adverse Events (CTCAE), the proportion of patients with pneumocystis jiroveci (PJP) infection, hospitalization, and treatment discontinuation rate. Randomized clinical trials, retrospective and prospective observational studies, and case series were included. Specific publication formats or avenues were excluded, including case reports, letters, opinion essays, commentaries, editorials, book chapters, and animal studies.

### 2.2. Literature Search

A research specialist (RS) designed and executed an electronic literature search. Publications on Medline; EMBASE; and Cochrane Central Register of Controlled Trials from 1946, 1947, and 2022, respectively, to April 2022 were included. To ensure we had a comprehensive search strategy, abstracts from the American Society of Clinical Oncology (ASCO), European Society of Medical Oncology (ESMO), American Association of Cancer Research (AACR), Society of Neuro-oncology (SNO), and European Association of Neuro-oncology (EANO) meetings published from 2009 to April 2022 were also included. The search strategy encompassed the terms "glioma", "oligodendroglioma", "oligoastrocytoma", "glioblastoma", "temozolomide", "chronotherapy", and their derivatives. A secondary, manual search of studies published from April 2022 to August 2022 was conducted on 30 August 2022. Only citations available in English were included. The protocol is registered within the Open Science Framework (OSF) database (DOI:10.17605/OSF.IO/KFWBJ), and the study is reported in compliance with the Preferred Reporting Items for Systematic Reviews and Meta-Analyses (PRISMA) 2020 statement [16].

### 2.3. Study Selection Process

Both stage 1 screening of titles and abstracts and stage 2 screening of eligible full text were conducted by two independent reviewers (JJ and BA). Any discordant decisions were arbitrated by a third reviewer (TN). Studies excluded after stage 1 and 2 screening with their reason for exclusion were recorded (Supplementary Tables S1 and S2). The study screening and selection process is outlined using a flow diagram (Figure 1).

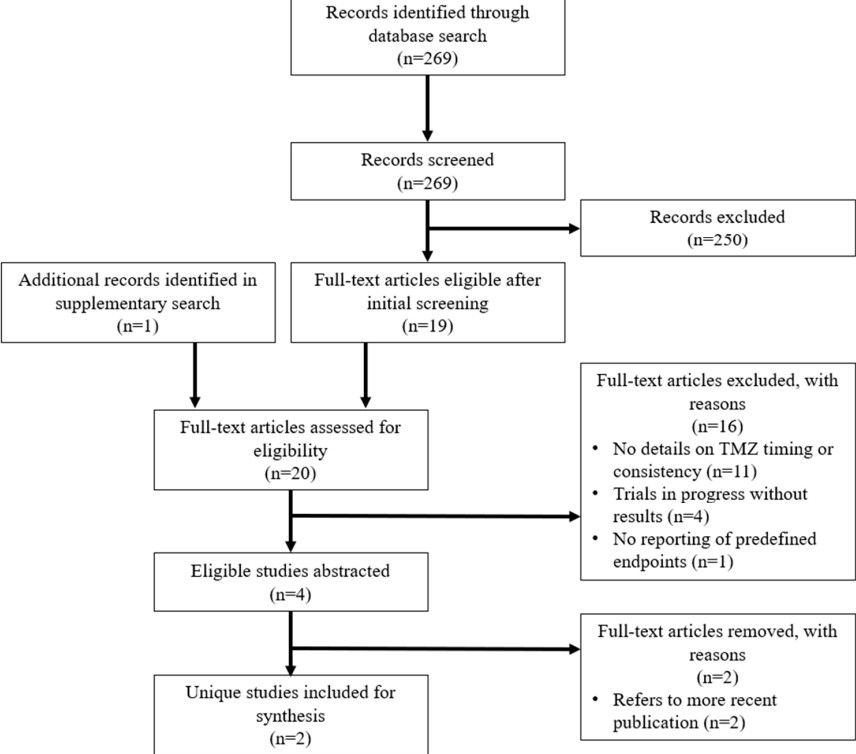

**Figure 1.** Flowchart of study screening and inclusion for abstraction.

*2.4. Data Collection and Risk-of-Bias Assessment*

All the eligible studies were abstracted for relevant study characteristics and outcomes, as noted above. Outcome data was taken from the most recent publication, whereas data from an older publication or abstract could be used as a supplement if they were missing from the most recent publication. Data from an abstract that was published after the most recent peer-reviewed full-text publication were also included if one of the pre-defined clinical outcomes was updated after a longer-term follow-up.

Using a standardized abstraction form, two independent abstractors (JJ and BA) carried out data abstraction. Studies were further assessed for risk of bias by independent reviewers (JJ and BA) using the Cochrane Collaboration's risk-of-bias tool for randomized studies [17] and the Newcastle-Ottawa quality assessment form for non-randomized studies [18]. Discrepancies were resolved with a third reviewer (TN). Findings were summarized narratively and in Table 1.

**Table 1.** Key study characteristics studies included for synthesis.

| Author | Year | Study Design | Country | Center | Span | Number of GBM Patients | Total Sample Size | Study Aim | Reported Outcomes |
|---|---|---|---|---|---|---|---|---|---|
| Damato et al. [19] | 2021 | Retrospective cohort | USA | Washington University School of Medicine | 2010–2018 | 166 | 166 | Evaluate the timing of adjuvant TMZ chronotherapy on GBM patient survival | OS, OS-12 |
| Damato et al. [20] | 2022 | Prospective feasibility | USA | Washington University School of Medicine | 2016–2020 | 21 | 35 | Feasibility and clinical impact of TMZ administration in the morning versus evening | OS, compliance, clinical and hematologic adverse effects, QoL |

## 3. Results

*3.1. Available Evidence*

Using the pre-defined search strategy, 269 publications were deemed potentially eligible. During first-stage screening, 250 publications were excluded because they did not describe the schedule (i.e., time-of-day) of TMZ administration. Reasons for exclusion are detailed in Supplement Table S1. Three were identified as potentially relevant during the second-stage screening of 19 articles. The updated manual search yielded one other publication. After reviewing all the candidate full-text manuscripts, four citations were eligible per the inclusion and exclusion criteria [19–22], listed in Table 1; and underwent complete abstraction. The details of article exclusion during second-stage screening are recorded in Supplement Table S2.

Throughout abstraction, it became apparent that two of the eligible conference abstracts [21,22] and one full-text manuscript reported on the same prospective study by Damato et al. (2022), as they referenced the same National Clinical Trial (NCT) number [20]. Therefore, only the recently published, comprehensive article was included for synthesis. Ultimately, two unique studies and articles were included for review. A flowchart of study selection per the PRISMA guidelines is illustrated in Figure 1.

*3.2. Study Characteristics*

All four eligible abstracts [19–22] were published by the neuro-oncology group at Washington University School of Medicine, United States (Table 1). However, only two articles were included for synthesis, as three publications originated from the same study at various stages of completion. Two were published as conference abstracts [21,22], and one was a recently published manuscript [20]. Only this full manuscript, which reported a

single-center prospective, randomized study (n = 35) comparing feasibility and adverse effects in two cohorts defined by the timing of TMZ administration, was included in the review synthesis. The remaining fourth article referred to a retrospective study (n = 166) aiming to compare the survival between GBM patients receiving adjuvant TMZ in the morning or evening.

The retrospective cohort study was categorized as good quality based on the Newcastle-Ottawa scale [18], while the prospective study was classified as being at a high risk for bias per Cochrane's risk-of-bias tool [17] (Table 2).

**Table 2.** Risk-of-bias assessment of all eligible studies.

| Newcastle-Ottawa Quality Assessment Scale for Cohort Studies | | | | | | |
|---|---|---|---|---|---|---|
| **Author** | **Year** | **Selection** | **Comparability** | **Outcome** | **Overall AHRQ Standard** | |
| Damato et al. [19] | 2021 | 3 stars | 0 stars | 2 stars | Good quality | |

| Cochrane Risk-of-Bias Tool for Randomized Trials | | | | | | | |
|---|---|---|---|---|---|---|---|
| **Author** | **Year** | **Selection** | **Comparability** | **Blinding Participants** | **Blinding Assessors** | **Complete Outcome Data** | **Selective Reporting** | **Overall Risk Judgment** |
| Damato et al. [20] | 2022 | Low | High | High | High | Low | Low | High |

*3.3. Efficacy and Adverse Effect Outcomes*

A summary of the reported outcomes, and results of survival and toxicity between groups are reported in Tables 3 and 4 respectively. The retrospective study [19] compared survival outcomes between patients who received adjuvant TMZ in the morning and evening. This was a single-center study, and the intervention of morning versus evening TMZ was clustered by physician practice; three physicians routinely administered TMZ in the morning (n = 89), and the remaining oncologist administered TMZ routinely in the evening (n = 77). The reported median OS in the morning and evening TMZ administration groups were 1.43 years vs. 1.13 years, respectively, suggesting that the morning administration of TMZ may improve clinical outcomes. Characteristics between groups were balanced except for the Karnofsky performance status (KPS), of which the evening cohort had a statistically significantly higher proportion of patients ≥ 80. Isocitrate dehydrogenase (IDH) wild-type status was reported in 93.5% of the evening group vs. 62.9% in the morning group (the remaining 37.1% missing IDH status). MGMT methylation status, the extent of surgical resection, and prior chemotherapy and radiation treatment were comparable between groups.

**Table 3.** Overview of outcomes reported in all eligible studies (OS: overall survival; PFS: progression-free survival; OS-12: overall survival at 12 months; GI: gastrointestinal; QoL: quality of life; X: reported; -: unreported). * Campian et al. [21] and Atluri et al. [22] report the pre-publication results of this completed study.

| Author | Year | OS | PFS | OS-12 | Heme Toxicity | GI Toxicity | Adherence | QoL |
|---|---|---|---|---|---|---|---|---|
| Damato et al. [19] | 2021 | X | X | X | X | - | X | X |
| Damato et al. [20] * | 2022 | - | - | | X | X | X | X |

**Table 4.** Reported survival and toxicity outcomes ** in all studies included for synthesis.

| Author | Year | Time | OS (y) | PFS (y) | OS-12 (%) | ORR (%) | Neutropenia (%) | | Thrombocytopenia (%) | | Anemia (%) | | Lymphopenia (%) | | AST | | ALT | | Nausea | | Vomiting | |
|---|---|---|---|---|---|---|---|---|---|---|---|---|---|---|---|---|---|---|---|---|---|---|
| | | | | | | | CTCAE Grade | | | | | | | | | | | | | | | |
| | | | | | | | 1/2 | 3/4 | 1/2 | 3/4 | 1/2 | 3/4 | 1/2 | 3/4 | 1/2 | 3/4 | 1/2 | 3/4 | 1/2 | 3/4 | 1/2 | 3/4 |
| Damato et al. [19] * | 2021 | AM | 1.43 | - | NR | - | - | - | - | - | - | - | - | - | - | - | - | - | - | - | - | - |
| | | PM | 1.13 | - | NR | - | - | - | - | - | - | - | - | - | - | - | - | - | - | - | - | - |
| Damato et al. [20] * | 2022 | AM | NR | NR | - | - | 0 | 0 | 10.0 | 0 | 15.0 | 0 | 5.0 | 0 | 0 | 5.0 | 0 | 5.0 | 25.0 | 5.0 | 15.0 | 0 |
| | | PM | NR | NR | - | - | 5.0 | 0 | 21.0 | 0 | 21.1 | 0 | 15.8 | 0 | 0 | 0 | 5.3 | 0 | 47.4 | 0 | 36.8 | 0 |

(OS: overall survival; PFS: progression-free survival; OS-12: overall survival at 12 months; ORR: objective response rate; AST: aspartate transaminase elevation; ALT: alanine transaminase elevation). * Campian et al. [21] and Atluri et al. [22] report the pre-publication results of this completed study. ** PJP incidence, hospitalization rate, and discontinuation rate were not described in either study.

The prospective randomized feasibility study that released preliminary results by Campian et al. [21]; then, Atluri et al. [22]; and finally, published as a peer-reviewed manuscript by Damato et al. [20] reported no significant difference in OS or PFS between the morning and evening administration groups. However, the sample size was too small (n = 35) for survival comparisons. Neither study reported OS or PFS at 6, 9, and 12 months; or the objective response rate.

Toxicity outcomes were only reported in the abstracts and paper from the prospective study. The conference abstracts indicated a higher incidence and severity of toxicity in the AM TMZ cohort. Specifically, Campian et al. reported a higher incidence and grade of hematologic toxicity, including thrombocytopenia, anemia, neutropenia, and lymphopenia in the morning compared to the evening cohort [21]. Accordingly, Atluri et al. reported worsened lymphocyte counts, although this was not statistically significant [22]. In the final manuscript, Damato et al. (2022) identified a higher overall incidence of treatment-emergent adverse events (TEAEs) in the evening cohort, but reported more cases of high-grade TEAEs in the morning cohort (n = 4) than in the evening cohort (n = 1) [20]. None of the studies reported incidence or rates of PJP infection, hospitalizations, or treatment discontinuation.

## 4. Discussion

There have been few advances in treating gliomas, especially for high-grade gliomas. The standard of care consisting of maximal surgical resection, adjuvant chemoradiation, and chemotherapy with temozolomide confers a survival advantage in the span of months compared to radiation therapy alone. There has been an urgent need to optimize survival for this challenging disease.

Chronotherapy proposes adjusting medication timing could enhance outcomes and maximize benefits based on pharmacokinetic and circadian principles. In the oncologic context, chronotherapy may refer to the timed administration of cytotoxic therapies that correlates with predictable oscillatory periods of vulnerability via mechanisms including, but not limited to, DNA repair, cell cycle progression, cytotoxic metabolism, and immune activity [21]. The hypothetical advantage would be maximizing chemotherapy-induced tumor death, minimizing collateral injury, improving survival, and decreasing adverse effects.

The circadian rhythm's role in malignant pathogenesis and interaction with cytotoxic agents is a subject of enormous depth and active study. There is limited, but favorable evidence at the clinical level. In metastatic colorectal cancer, chronotherapeutically administered doublet oxaliplatin and fluorouracil (5-FU) with folinic acid (FOLFOX) versus constant-rate infusion FOLFOX produced a superior objective response rate (51% versus 29%) and decreased grade 4 toxic events, even though there was no detected difference in survival. In this instance, the authors administered 5-FU in the early morning, a period of homeostasis, with decreased DNA synthesis and 5-FU catabolism, which conceivably conferred improved tolerability [14]. In non-B-cell acute lymphoblastic leukemia, evening administration of methotrexate and 6-mercaptopurine was associated with an increased probability of event-free survival compared to the morning schedule [13]. Since then, other studies have reported fewer adverse effects with chronotherapeutically administered cisplatin in non-small cell lung cancer [23] and ovarian cancer [24], and irinotecan in colorectal cancer [25].

Chronotherapy of TMZ in high-grade glioma is an appealing strategy since there is a low bar for improvement in this population, and the implementation of a change in the timing of oral medication is so simple, without any perceived implications for resource utilization (as opposed to intravenously administered systemic therapy). Through a systematic and rigorous literature search, we confirmed that only one group had published clinical data on the outcomes of chronotherapeutically administered TMZ in glioma. Trends of increased survival and high-grade toxicity associated with morning TMZ administration suggest consistency with the pre-clinical findings of increased glioma cell cytotoxicity at the peak of *bmal1* expression and the nadir of MGMT expression—both of which seemed to demonstrate important circadian rhythms.

Ultimately based on this systematic review, there is a clear need for further data to ascertain the value of TMZ chronotherapy. A search of clinicaltrials.gov reveals no other completed or ongoing relevant studies in the glioma population. Fortunately, Damato et al. [19] had already demonstrated that 95% of their participants adhered to the randomly assigned treatment schedule. Accordingly, a larger prospective randomized study investigating overall survival, health-related quality of life (HR-QoL), and treatment-related toxicity should be feasible.

Although TMZ chronotherapy could provide immediate benefits for patients with glioma, we do not have definitive clinical data yet to support a unanimous change to morning TMZ administration. However, parts of the neuro-oncology community have already started prescribing TMZ in the morning based on the preliminary clinical data presented in this review. Clearly, there is an urgent need to confirm the absolute value of TMZ chronotherapy, or lack thereof, so that patients can continue to receive optimal and equitable medical care.

Given the perceived minimal risk and presence of the clinical equipoise concerning the timing of TMZ administration, the impact of TMZ chronotherapy on survival, toxicity, and HR-QoL can easily be assessed using a pragmatic clinical trial design. The main advantages of a pragmatic trial design include a study cohort that is more representative of the real-world patient population, lower cost of clinical trial operation due to much fewer regulatory requirements (and study-mandated procedures), and faster time to study accrual and completion due to the broader eligibility criteria and the minimal risk nature of the intervention. The Rethinking Clinical Trials (REaCT) group is one example of a pragmatic trials program with a successful history of conducting prospective trials that compare commonly used treatments within a validated, pragmatic, and efficient trial methodology [26,27]. We advocate using the REaCT framework or similar approaches to swiftly develop a multicenter trial to address the role of TMZ chronotherapy.

While we wait for novel and effective systemic therapies from large, mostly industry-sponsored studies, we must remember that very few breakthroughs have occurred in the last two decades and that second-line therapies used at the time of disease recurrence remain quite limited. Therefore, it has become more important to ensure that we are fully optimizing the use of TMZ as it is the only drug regimen supported by level-one evidence in this population. Ultimately, we are hopeful that TMZ chronotherapy will serve as a practical and yet under-investigated avenue to improve the care of patients with this challenging disease.

**Supplementary Materials:** The following are available online at https://www.mdpi.com/article/10.3390/curroncol30020147/s1, Table S1: List of studies excluded during 1st stage screening, with reason for exclusion; Table S2: List of studies excluded during 2nd stage screening, with reason for exclusion.

**Author Contributions:** All the authors contributed to the study conception and design. J.L.J., B.A. and T.L.N. performed the material preparation, data collection, analysis, and interpretation. J.L.J. and T.L.N. wrote the first draft of the manuscript. All the authors commented on previous versions of the manuscript. All authors have read and agreed to the published version of the manuscript.

**Funding:** The authors declare that no funds, grants, or other support were received for the preparation of this manuscript.

**Acknowledgments:** The authors would like to acknowledge Risa Shorr, the Research Specialist and Librarian at Ottawa Hospital, for performing a literature review for the current study.

**Conflicts of Interest:** The authors declare no conflict of interest.

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
