# Peer review of "Temozolomide Chronotherapy in Glioma: A Systematic Review"

_curroncol, doi:10.3390/curroncol30020147_

Round 1

Reviewer 1 Report

The aim of this literature review is to put together all available information on the timing of treatment of glioblastoma multiforme (GBM) with temozolomide (TMZ), and to discuss efficacy and safety reported as a function of the timing. In addition the article highlights the mechanism that is supposed to have an influence on the efficacy of TMZ (and of other chemotherapeutics) as a function of the time of the day.

This is the first time that such a review is undertaken; it is of high interest and an important, relevant contribution to aspects of chronotherapy of tumours, specifically for the treatment of GBM with TMZ. The review will likely support future systematic chronotherapeutic studies and also physicians in their treatment decision.

The available literature has been reviewed carefully and critically; the manuscript is clearly and well written. Tables are comprehensive.

As there is only one entry in the row “Comment for “Other” (Supplement Tables), it is recommended to skip this column and to refer to the publication [Wen, P. Y. (2010)] in form of an asterisk and a foot note (“Article inaccessible”) instead.

The article does well fit the scope of the Journal.

It is recommended to publish this review.

Author Response

We thank reviewer 1 for reviewing our manuscript and providing valuable feedback. 

Reviewer 1: The aim of this literature review is to put together all available information on the timing of treatment of glioblastoma multiforme (GBM) with temozolomide (TMZ), and to discuss efficacy and safety reported as a function of the timing. In addition the article highlights the mechanism that is supposed to have an influence on the efficacy of TMZ (and of other chemotherapeutics) as a function of the time of the day. This is the first time that such a review is undertaken; it is of high interest and an important, relevant contribution to aspects of chronotherapy of tumours, specifically for the treatment of GBM with TMZ. The review will likely support future systematic chronotherapeutic studies and also physicians in their treatment decision. The available literature has been reviewed carefully and critically; the manuscript is clearly and well written. Tables are comprehensive.

Response: We thank Reviewer 1 for reviewing our manuscript and providing valuable feedback.

As there is only one entry in the row “Comment for “Other” (Supplement Tables), it is recommended to skip this column and to refer to the publication [Wen, P. Y. (2010)] in form of an asterisk and a foot note (“Article inaccessible”) instead. The article does well fit the scope of the Journal. It is recommended to publish this review.

Response: We thank Reviewer 1 for this feedback, and we agree. Accordingly, the supplement document was revised.

Reviewer 2 Report

1The authors conducted systematic review on an understudied concept, highlighting the need for more studies in this direction. The following minor revision is required.

1. Reference required for “….MGMT ....5-fold variation oscillation with a peak during night-time and low light.”  in Line 70

Author Response

We thank Reviewer 2 for reviewing our manuscript and providing valuable feedback.

Reviewer 2:  The authors conducted systematic review on an understudied concept, highlighting the need for more studies in this direction. The following minor revision is required. Reference required for “….MGMT ....5-fold variation oscillation with a peak during night-time and low light.”  in Line 70

Response: We thank Reviewer 2 for this great suggestion, and we agree. We have inserted an accompanying citation (Martineau-Pivoteau et al., 1996) that identified MGMT activity varying 5-fold across a 24-hour time period. For clarification, we also added a small comment on MGMT activity (in contrast to protein expression) as well as determination within the mouse liver.

Reviewer 3 Report

The authors provide a systematic review of the efficacy, toxicity and health-related quality of life of temozolomide chronotherapy. The trends in increased survival and toxicity associated with morning temozolomide administration in clinical data are consistent with preclinical findings. This is an important topic for which a comprehensive review is scarce in the literature. The authors summarized the most recent and relevant papers on the subject. Although there are several typos and the narrative is a kind of fragmentary, I just abide by scientific soundness. The text is well written and very easy to read and follow it. I would like to offer the following minor points for consideration by the authors towards the improvement of the manuscript:

1-  Line 11 Authors mentioned that “Temozolomide is the only drug approved for first-line treatment.” . I would suggest to please confirm this.

2- Please elaborate on the exclusion criteria in main manuscript(for example book chapters, books, case reports, editorial letters, review articles, retrospective studies, single-arm studies, and opinion papers; animal studies; studies not in English). Please check the accuracy of number in study selection (252+19=271)

3-Please specify abbreviations where they are first mentioned in the text. Some examples include: bmal1, per2

Author Response

We thank Reviewer 3 for reviewing our manuscript and providing valuable feedback.

Reviewer 3: The authors provide a systematic review of the efficacy, toxicity and health-related quality of life of temozolomide chronotherapy. The trends in increased survival and toxicity associated with morning temozolomide administration in clinical data are consistent with preclinical findings. This is an important topic for which a comprehensive review is scarce in the literature. The authors summarized the most recent and relevant papers on the subject. Although there are several typos and the narrative is a kind of fragmentary, I just abide by scientific soundness. The text is well written and very easy to read and follow it. I would like to offer the following minor points for consideration by the authors towards the improvement of the manuscript:

Response: We thank Reviewer 3 for the in-depth analysis.

Reviewer 3: 1-  Line 11 Authors mentioned that “Temozolomide is the only drug approved for first-line treatment.” . I would suggest to please confirm this.

Response: We thank Reviewer 3 for pointing this out, and we agree. We have clarified that it is the sole, preferred first-line therapy for GBM specifically and inserted a citation to the National Comprehensive Cancer Network (NCCN) guideline on CNS treatment. (Introduction, page 1, lines 42-43).

Reviewer 3:  2- Please elaborate on the exclusion criteria in main manuscript(for example book chapters, books, case reports, editorial letters, review articles, retrospective studies, single-arm studies, and opinion papers; animal studies; studies not in English). Please check the accuracy of number in study selection (252+19=271)

Response: We thank Reviewer 3 for this feedback, and we agree. The manuscript was revised accordingly (Methods, page 3, lines 106-109).

Reviewer 3:  3-Please specify abbreviations where they are first mentioned in the text. Some examples include: bmal1, per2

Response: We thank Reviewer 3 for pointing this out, and we agree. We have defined and clarified all acronyms / genes that were mentioned for the first time in the manuscript.  These include: Basic Helix-Loop-Helix ARNT Like 1 (bmal1) and Period Circadian Regulator 2 (per2), pneumocystis jiroveci (PJP), Preferred Reporting Items for Systematic Reviews and Meta-Analyses (PRISMA), National Clinical Trial (NCT), and isocitrate dehydrogenase (IDH).

Additional changes made:

Page 1, line 22: “The descriptive analysis was followed due to limited data.” was reworded to “A descriptive analysis was conducted due to limited data.”

Line 50: added “…and is an externally worn device…”

Line 76-77: “Chronotherapy has already been demonstrated in leukemia and colorectal cancer” was changed to “Chronotherapy has already shown some promise in leukemia and colorectal cancer.”

Line 130: deleted “were”

Line 175: deleted “a”